# A Simulation of the Spatial Expansion Process of Shrinking Cities Based on the Concept of Smart Shrinkage: A Case Study of the City of Baishan

Wancong Li [1], Hong Li [1], Feilong Hao [2], Zhiqiang Feng [3] and Shijun Wang [2,*]

1   College of Earth Sciences, Jilin University, Changchun 130061, China; liwc19@mails.jlu.edu.cn (W.L.); h_li@jlu.edu.cn (H.L.)
2   School of Geographical Sciences, Northeast Normal University, Changchun 130024, China; haofl587@nenu.edu.cn
3   School of GeoSciences, The University of Edinburgh, Edinburgh EH8 9XP, UK; zhiqiang.feng@ed.ac.uk
*   Correspondence: wangsj@nenu.edu.cn

**Abstract:** The coexistence of urban expansion and shrinkage in China has become increasingly apparent; therefore, the current strategic model of growth-oriented urban planning as the top-level design needs to be adjusted. This paper focuses on the city of Baishan, which is a typical shrinking city in China, and explores the feasibility of implementing the concept of smart shrinkage planning in shrinking cities in China by constructing a coupled PLUS-SD model. The results demonstrate the following conclusions: (1) The overall simulation of the coupled PLUS-SD model is superior to that of the PLUS model. In Baishan, the areas with the most changes in construction land will be located at the edges of the landforms by 2030. (2) Using the traditional planning scenario would only exacerbate the rate of construction land expansion in Baishan, deepening the incongruity between the city's population and construction land. (3) The smart shrinkage scenario will require strict control of the scale of construction land and optimization of the structure of the urban construction land, which would push the city in the direction of healthy and sustainable development. (4) The concept of smart shrinkage planning is a scientific and feasible plan for realizing the efficient and sustainable use of construction land in shrinking cities.

**Keywords:** shrinking city; smart shrinkage; coupled PLUS-SD model; multiple scenario simulations; Baishan

## 1. Introduction

Urban shrinkage is a widespread and complex phenomenon, as well as a popular topic in international urban research in recent years [1]. Since World War II, due to suburbanization and the large-scale movement of industries internationally and domestically, the numbers of industries and employed people in most industrial cities in developed countries have decreased, resulting in their development suffering a Waterloo-style decline [2]. These cities encompass some former hubs of industry, such as Youngstown, Liverpool, Detroit, Leipzig, and Berlin, and even Detroit has experienced urban financial bankruptcy [3]. In order to distinguish this from the negative phenomenon of urban decline caused by the loss of production factors due to the evolution of the industrial structure in the past, Huermann et al. [4] introduced the idea of urban shrinkage in a study conducted in 1988, which focused on population decline in the Drucker region of Germany. Since then, this concept has been widely applied. China is in a period of rapid urbanization; therefore, considering expansion and growth from the perspective of logical deduction seems to be the conventional progression pathway for the development of cities [5]. Nevertheless, China has experienced different levels of urban shrinkage in both quickly developing and relatively less developed regions [6]. Moreover, there has been a consistent increase in

the quantity of shrinking cities in China in recent years. Throughout China's nearly four decades of tremendous economic growth due to the implementation of reform and the opening up of policies, many cities have developed in the form of disordered expansion under the guidance of growth-oriented urban planning concepts, and the rate of population urbanization has been far slower than that of land urbanization [7]. In cities with growing populations, this growth planning philosophy based on the high rates of city growth, the current high rate of population growth, and the anticipation of sustained growth of the city's population in the future can still be justified. However, in cities that are already experiencing population shrinkage, such planning concepts are difficult to justify in any fundamental logical sense. The contradiction between the decreasing urban population and increasing construction land has also become a pressing issue for China's shrinking cities (Figure 1) [6].

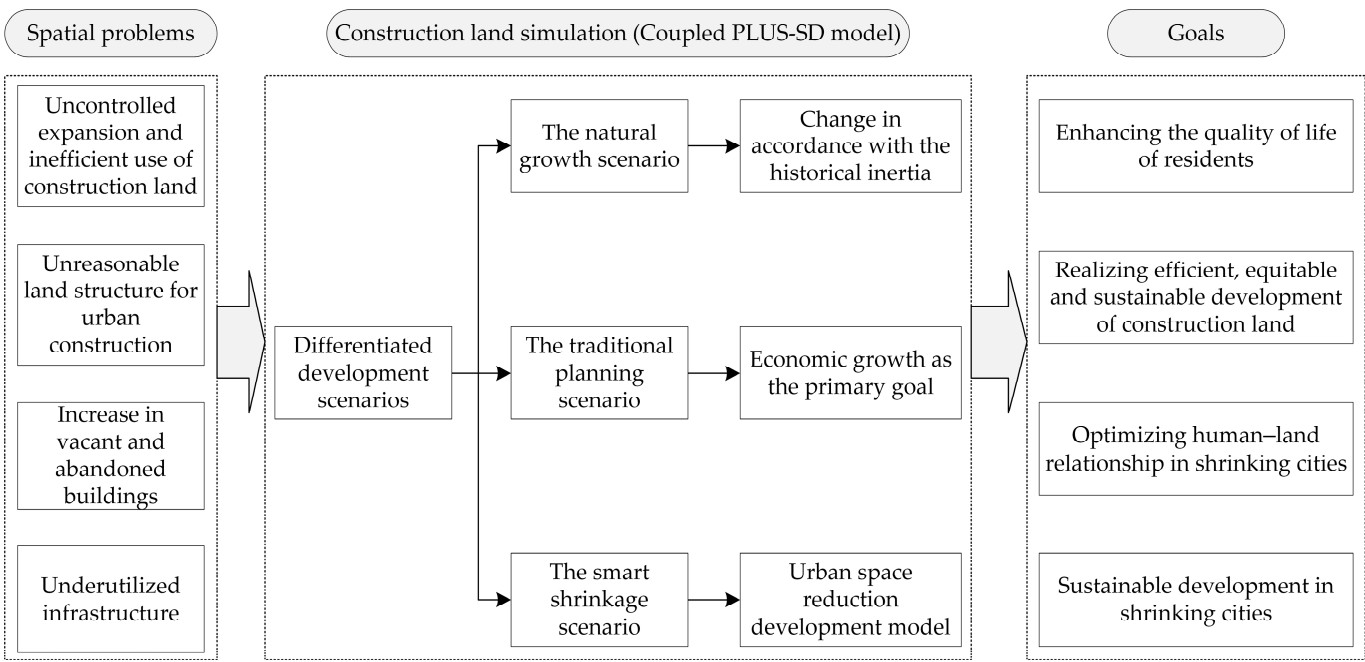

**Figure 1.** Analytical framework for simulating the spatial evolution process of a shrinking city.

Construction land simulation and optimization models are an important part of land science and are a fundamental way to guarantee that shrinking cities can sustainably use land resources. Optimizing the quantitative structure and spatial allocation of urban construction land under certain constraints is important and provides a tool for the rational allocation of construction land. With the development of spatial simulation and intelligent optimization technology, the theoretical basis and methodological systems for the optimal allocation of urban construction land are constantly being updated, but they can still be divided into two major categories: optimization of the quantitative structure and optimization of the spatial allocation of construction land. The former is mainly a rational allocation arrangement of the proportion of urban construction land, such as linear planning, multi-objective planning [8], Markov chains, gray prediction, system dynamics [9,10], and other models. The latter is a spatially rational layout of urban construction land, which mainly includes the conversion of land use and its effects (CLUE) model [11], the conversion of land use and its effects at small regional extent (CLUE-S) model [12], the cellular automata (CA) model [13,14], the multi-agent systems (MAS) model [15], the future land use simulation (FLUS) model [16], the patch-generating land use simulation (PLUS) model [17], and bionic intelligent algorithms inspired by the natural laws of the living world, such as genetic algorithms (GAs) [18], ant colony optimization (ACO) [19], and particle swarm optimization (PSO) [20]. However, the established construction land quantity and structure optimization model mainly adjust the quantity allocation of urban construction land and

cannot solve the spatial distribution problem. The spatial distribution optimization model focuses on reflecting the spatial distribution and evolution of urban construction land, but there are limitations in the selection of the indicator factors. Therefore, constructing an interactive relationship between a quantitative structure and spatial layout optimization and determining the interactive validation mechanism between the quantitative structure and spatial simulation prediction are urgent problems that need to be solved in current spatial optimization decision-making regarding construction land use.

In view of this, Baishan, a representative shrinking city in Northeastern China, is taken as a case study in this paper, and the coupled patch-generating land use simulation-system dynamics (PLUS-SD) model is utilized, which can give full play to the spatial expression advantages of the PLUS model and the complex system simulation of the SD model. In addition, this study constructs three development scenarios for the future development plan of Baishan: natural growth, traditional planning, and smart shrinkage [21,22]. Subsequently, the feasibility of implementing the concept of smart shrinkage planning in China's shrinking cities is explored by comparing the characteristics of the spatial pattern and the differences in the utilization efficiency of construction land in Baishan under three different development scenarios (Figure 1). The results of this study provide a conceptual structure and practical evidence for improving the connection between humans and the land in shrinking cities and can provide a basis for the governments of China's shrinking cities to formulate relevant development plans. In addition, this paper incorporates big data (such as POI data) into the system comprising the driving factors of the evolution of construction land in shrinking cities to enhance the rationality of construction land simulation [23,24]. This makes the research on the optimal allocation of construction land in shrinking cities more scientific and informative. Furthermore, optimization simulation research on shrinking construction land based on the two perspectives of construction land and its internal structure can effectively make up for the shortcomings of research based on only a single perspective [25,26] and provides a reference basis for the accurate simulation of shrinking urban construction land.

## 2. Theoretical Foundation

As the number of internationally shrinking cities continues to grow, developed countries are proposing different governance policies based on their own political systems, policy environments, and administrative systems [27,28]. For example, to varying degrees, Germany's reconstruction of the Eastern Regions program [29,30], Japan's decentralization policy, and the United Kingdom's diversified urban regeneration policy [2] have served as references for economic, industrial, and social development strategies for China's shrinking cities. However, in the face of China's strong government-led market economy, in which the government monopolizes the primary market for urban land use and expropriation, smart shrinkage, which many scholars have proposed as a new concept and paradigm for urban planning, may be more appropriate for China's governance of shrinking cities. The concept of smart shrinkage planning includes taking a positive, developmental attitude towards the loss of the urban demographic, improving the efficiency of urban construction land use through the rational exit and optimal reorganization of resources, and achieving sustainable urban development [31–33].

In recent years, through analysis of the results of the implementation of smart shrinkage planning in various cities in the United States, scholars have found that the implementation of smart shrinkage planning in the following three aspects of the implementation of the city has a greater impact. First, guided by the principle of intelligent reduction planning, local governments can close down some redundant public facilities and municipal utilities, thus reducing the government's financial expenditures [31]. Second, by means of land expropriation, the population can be relocated from areas with low population densities in shrinking cities to high-density areas [34], and the former relocation areas can be transformed into open spaces or green spaces and squares, the infrastructure of high-density areas can be improved, and the quality of life of the inhabitants can be enhanced in a

multi-dimensional manner [35]. Third, abandoned and unused buildings may contribute to a rise in the crime rate within the local community, forcing residents to move out [36,37]. However, the renovation of abandoned and unused buildings not only improves public security and the public environment in a shrinking city, but also increases the value of the buildings in the neighborhood by 30% [38].

In terms of empirical evidence of planning for smart shrinkage, the city of Youngstown in northeastern Ohio, located in the northern part of the Rust Belt, which has the highest concentration of shrinking cities in the United States, is a representative shrinking city. The development of Youngstown has gone through five phases: emergence (1797–1980), prosperity (1840–1940), stagnation (1940–1960), shrinkage (1960–2010), and governance (2010–present). At the beginning of the city's development, Youngstown was rich in mineral resources and developed an industrial chain dominated by coal resource mining and processing, which attracted a large number of people to the city. This caused the city's population to surge from 654 in 1840 to 167,720 in 1940, resulting in the city's rapid development. However, with the decline of the demand for steel in the global and local markets and the city's location disadvantage, the city's economic development entered a state of stagnation. This led to stagnation in the size of the city's population, which remained at around 170,000 during 1940–1960 (Figure 2a). The subsequent oil crises, de-industrialization, suburbanization, white flight, and rise of the Sunbelt have led to notable declines in the city's population, with a rate of population decline of more than 15% per decade between 1960 and 2010. While the number of dwellings in the city decreased by 35.03% during this period, the city's residential vacancy rate continued to increase from 4.66% to 18.97% as a result of the continued exodus of the population, and it exceeded the Ohio average in 1980 and the U.S. average in 2000 (Figure 2b). The number of unoccupied residential lots and crime rate continued to increase, and the city entered a period of absolute shrinkage.

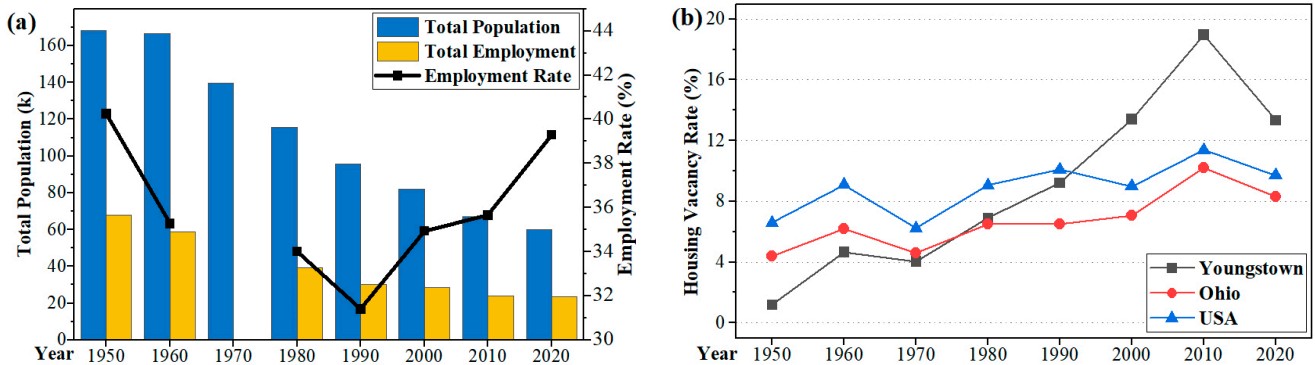

**Figure 2.** (**a**) Trends of Youngstown's population during 1950–2020; (**b**) Analysis of residential vacancy rates in Youngstown, Ohio and the USA during 1950–2020 (data from the U.S. decennial census for the corresponding years).

However, since the implementation of smart shrinkage-oriented urban planning in Youngstown, the rate of decline of the city's population has slowed down (around 10%), 39.16% of the unused and abandoned residential land has been cleared, and the city's residential vacancy rate has decreased significantly (13.33%). This also serves as favorable proof of the effectiveness of the governance of the urban space in this shrinking city by implementing the concept of smart shrinkage planning. The city of Youngstown took the following major steps in the 2010 Citywide Comprehensive Plan: (1) it reduced the residential land by 30%; (2) it reduced the commercial land by 16% on the basis of retaining the current main commercial corridor; (3) it doubled the educational land; (4) it developed industrial green land, reduced the size of the original heavy industry and light industry in the city, and developed environmentally friendly industries; and (5) it doubled the amount

of land set aside for open space in the city (Figure 3). Therefore, implementing smart growth is a viable strategic tool for urban redevelopment in shrinking cities.

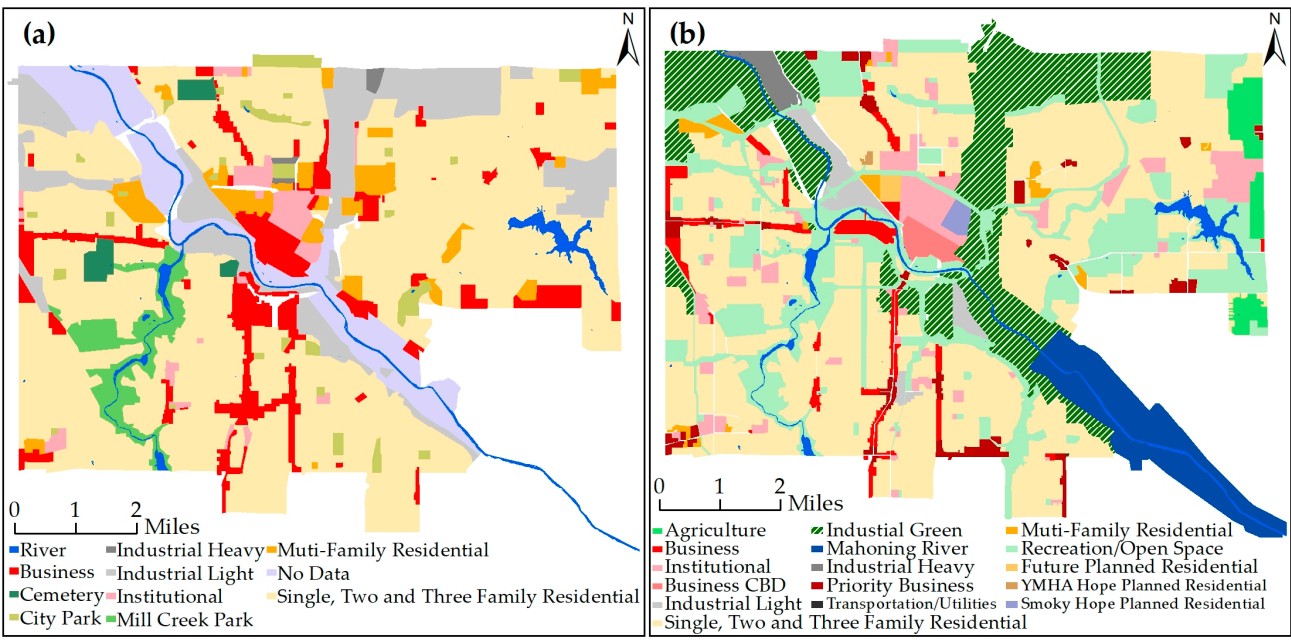

**Figure 3.** (**a**) Youngstown's land use status in 2010; (**b**) the Youngstown 2010 Citywide Comprehensive Plan's planned land use (the figures were drawn based on the planning).

## 3. Data and Methods

### 3.1. Study Area

#### 3.1.1. Geographic Location

Northeastern China is a region with a long history as an industrial center, and after the founding of the country's economy, it also took the lead in the development of a relatively independent and complete economic zone. In the early years of the establishment of China, Northeastern China provided strong economic support. Nevertheless, in recent years, as a result of the absence of geographic location benefits and excessive resource extraction, this region has been characterized by a low rate of economic development, continuous loss in population, uncontrolled expansion of land for construction, and destruction of habitats and the environment [39]. It has also become the region with the highest concentration of shrinking cities in China. Baishan is situated in the southeastern part of Northeastern China, in the hinterland of Changbai Mountains, which are rich in mineral resources, making it a resource-based city and an old industrial city (Figure 4). China's resource cities and old industrial cities are more likely to evolve to shrinking cities due to their historical development problems, which are characterized by a single industrial structure, dependence on industrial paths, and greater shrinkage pressure. In this paper, Baishan was selected as a case study. Its optimal allocation plan of construction land is a microcosm of resource-based shrinking cities and old industrial shrinking cities and enhances the quality of construction land utilization in those cities. In addition, the shrinkage of districts in Chinese cities is more closely aligned with the worldwide characteristics of shrinking cities [40]; therefore, the districts in Baishan were selected as research units in this paper.

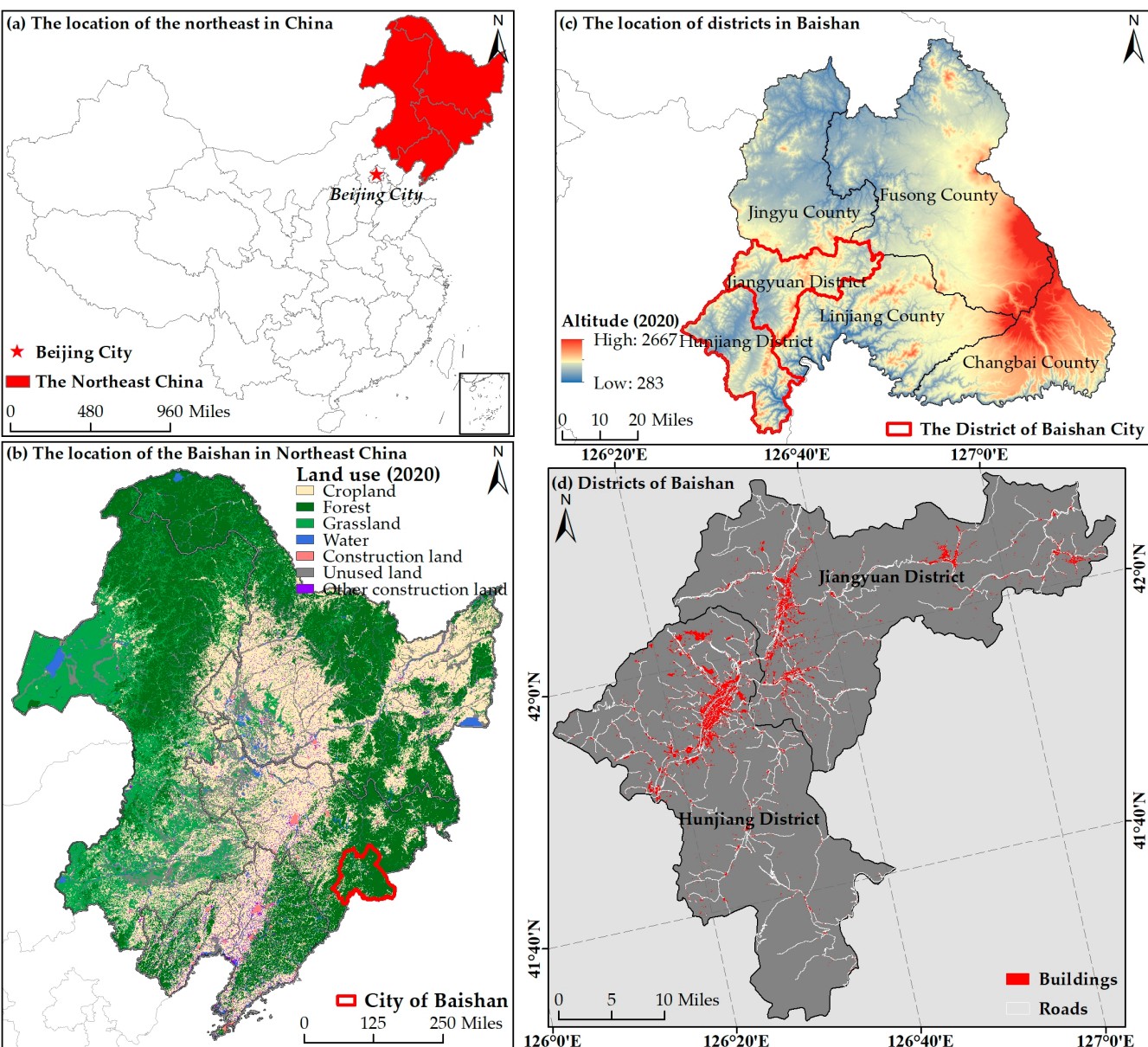

**Figure 4.** Geographic location of the study area in China.

### 3.1.2. Population Profile

From 2000 to 2010, the investment drive under the Northeast Revitalization Strategy, the support of its own factor resource endowment, and the movement of people between urban and rural areas under rapid urbanization made up for the fact that the economic development of Baishan lacked an internal drive, and the population of the urban area of the city still increased slowly by 24,100 people. From 2010 to 2020, with the modification of the national economic development pattern, the Northeast Revitalization Strategy entered a weak period, and the city's industrial transformation is still lagging behind other cities in China (in 2020, Baishan had only formed the "inverted triangle" development pattern of the industrial output ratio). In addition, in the process of de-industrialization, it is difficult for the jobs provided by the tertiary industry to make up for the jobs lost as a result of the industrial decline, resulting in a decrease of 172,800 people in the city's urban resident population during this period, which is a shrinkage of 27.91%, making it the city with the most serious population loss in Northeastern China.

### 3.1.3. Land Use Profile

Despite the population decline, the construction land in the city's districts increased from 10.81 km² in 2000 to 41.33 km² in 2020, with an exponential increase in the amount of new construction land during these two decades (Table 1), causing the city to experience a contradictory phenomenon—that is, a decrease in population and the expansion of construction land. Most of the new construction land is residential land. This phenomenon of the over-development of real estate has led to the increasingly serious problem of vacant urban buildings and land, and the livability of the city has been declining.

**Table 1.** Changes in the construction land and structure in Baishan.

| Year | Construction Land (km²) | Land Structure of Urban Construction (km²) | | | | | | |
|---|---|---|---|---|---|---|---|---|
| | | RL [1] | PUL [2] | IL [3] | WL [4] | RTL [5] | MUL [6] | GSPL [7] |
| 2000 | 10.81 | 10.04 | 1.56 | 3.24 | 0.46 | 1.85 | 2.20 | 1.15 |
| 2010 | 25.10 | 15.83 | 2.50 | 5.29 | 1.28 | 4.05 | 0.82 | 4.22 |
| 2020 | 41.33 | 18.39 | 4.04 | 6.88 | 0.94 | 6.39 | 1.05 | 5.47 |

[1] RL is residential land; [2] PUL is public utilities land; [3] IL is industrial land; [4] WL is warehousing land; [5] RTL is road traffic land; [6] MUL is municipal utility land; [7] GSPL is green space and plaza land.

Moreover, the city has continued to develop and expand in a disordered fashion, leading to a continuous loosening of the city form during the spatial expansion process over the past 20 years. Thus, the compactness decreased from 0.2717 in 2000 to 0.1303 in 2020 (Table 2). From 2010 to 2020, the level of intensive utilization of the construction land in Baishan exhibited a contracting trend, which mainly manifested in the double contraction of the utilization intensity and output efficiency. The former was mainly triggered by the population loss and construction land expansion, especially the increase in the area of residential land. The latter was caused by factors such as shrinking investment, the transfer of the scientific and technological innovation capacity, and the outward migration of new industries.

**Table 2.** Statistics of the utilization efficiency of the construction land in Baishan.

| Year | Compactness | Intensity | Input Level | Utilization Intensity | Output Benefits | Sustainability |
|---|---|---|---|---|---|---|
| 2000 | 0.2717 | 0.3039 | 0.0417 | 0.2153 | 0.0276 | 0.0193 |
| 2010 | 0.1510 | 0.3746 | 0.0492 | 0.1752 | 0.1136 | 0.0366 |

That is, under the combined effect of external environmental changes (the national economic pattern shifted from the northeast region to the southeast coastal zone) and internal transformation (the depletion of mineral resources and the ecological damage caused a lag in the transformation of the industrial structure and the lack of successor industries), Baishan's social and economic structure have exhibited a general decline. The city has exhibited phenomena such as industrial decline, population outflow, housing vacancies, and the inefficient utilization of construction land. These negative impacts are mutually reinforcing and causative factors, forming a vicious circle that causes the phenomenon of urban shrinkage in Baishan to gradually intensify. Therefore, in this paper, the districts in Baishan were selected as a case study to test the concept of smart shrinkage planning with typical representativeness and practical significance.

### 3.2. Data

The primary sources of the data utilized in this study are outlined below. (1) The *land use data* (30 m) were derived from decoded data (2000, 2010, and 2020) obtained from the Resource and Environmental Data Cloud Platform (http://www.resdc.cn (13 October 2021)) and are divided into seven land use categories: cropland, forest, grassland, water, construction land (i.e., urban land use), unused land, and other construction land (i.e., the total number of rural settlements and other construction land). (2) The population

data were obtained from the National Population Census (fifth, sixth, and seventh), and the socio-economic data were obtained from the China Urban Statistical Yearbook, the China Urban Construction Statistical Yearbook, and the statistical yearbooks and statistical bulletins of each province for the corresponding years. (3) The coupled PLUS-SD model land use simulation *driving data* are presented in Table 3. (4) In the coupled PLUS-SD model simulations, three types of rigid development restriction zones, namely, permanent basic farmland, nature reserves, and water systems, are used to constrain the disorderly development of construction land in the simulation process (Figure 5), which can better reflect the reality of the land use evolution in Baishan. In addition, according to the Baishan City Master Plan (2015–2030), parcels with relative heights of greater than 30 m and slopes of greater than 15° are also classified as restricted expansion areas for traditional planning scenarios.

**Table 3.** Information about driving factors for land use simulation in Baishan.

| Selection Levels | Type | Driving Factor | Data Description | Year | DS [1] |
|---|---|---|---|---|---|
| Natural environmental factors | Water resources | a. Distance from the river system | 30 m × 30 m raster | 2020 | d |
| | Land resources | b. Soil types | 227 subcategories | 2010 | a |
| | Vegetation resources | c. Annual vegetation index | NDVI | 2010/2020 | b |
| | Climatic conditions | d. Average annual precipitation | 100 m × 100 m raster | 2010/2020 | a |
| | | e. Average annual temperature | 100 m × 100 m raster | 2010/2020 | a |
| | Geologic disasters | f. Distance to geologic hazard sites | 7 disaster sites | 2019 | a |
| | Topographic conditions | g. Altitude | 90 m × 90 m raster | 2020 | b |
| | | h. Slope | 90 m × 90 m raster | 2020 | b |
| Socio-economic factors | Population | i. Population density (people/km$^2$) | 100 m × 100 m raster | 2010/2020 | c |
| | Economy | j. GDP (10,000 yuan/km$^2$) | 100 m × 100 m raster | 2010/2020 | a |
| Location condition factors | Traffic | k. Distance to the highway | 30 m × 30 m raster | 2019 | a |
| | | l. Distance to the national highway | 30 m × 30 m raster | 2019 | a |
| | | m. Distance to the provincial highway | 30 m × 30 m raster | 2019 | a |
| | | n. Distance to main urban roads | 30 m × 30 m raster | 2019 | a |
| | | o. Distance to the railroad | 30 m × 30 m raster | 2019 | a |
| | Facility accessibility | p. Distance to the seat of government | 30 m × 30 m raster | 2020 | e |
| | | q. Distance to shopping centers | 30 m × 30 m raster | 2020 | e |
| | | r. Distance to higher education institutions | 30 m × 30 m raster | 2020 | e |
| | | s. Distance to main hospitals | 30 m × 30 m raster | 2020 | e |
| | | t. Distance to the park land | 30 m × 30 m raster | 2020 | e |
| | | u. Distance to industrial parks | 30 m × 30 m raster | 2020 | e |

[1] DS stands for data sources. For data sources: a: Resource and Environment Science Data Center (http://www.resdc.cn (13 October 2021)); b: Geospatial Data Cloud (https://www.gscloud.cn (24 March 2023)); c: Open Spatial Demographic Data and Research (https://www.worldpop.org (24 March 2023)); d: Open Street Map (https://www.openstreetmap.org (24 March 2023)); e: Baidu Map API Crawler Data (http://map.baidu.com (24 March 2023)).

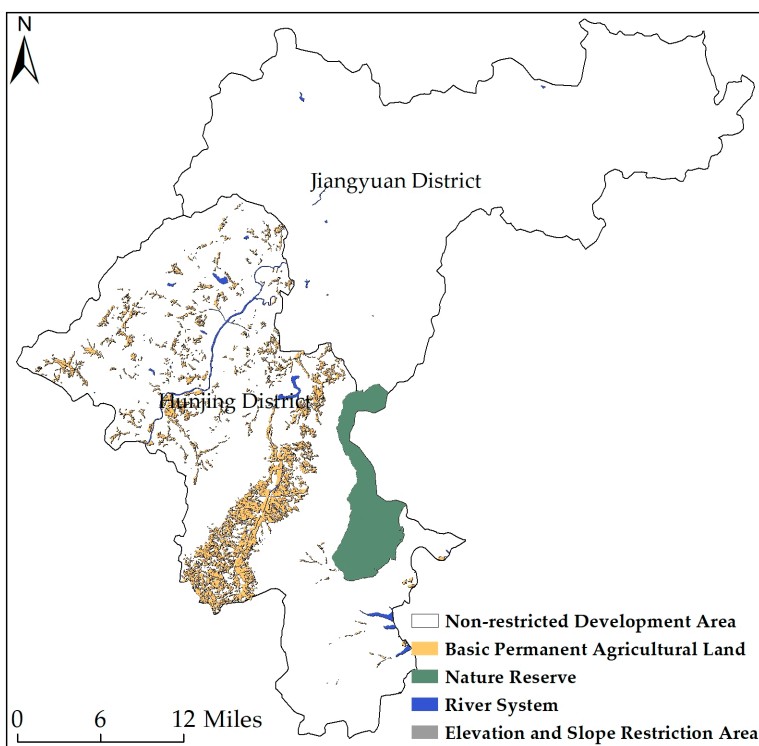

**Figure 5.** Restricted development areas in Baishan.

*3.3. Models and Methods*

3.3.1. Coupled PLUS-SD Model

The patch-generating land use simulation (PLUS) model is a land use change simulation model that utilizes patch data to effectively analyze the mechanisms behind land use change and to accurately simulate the development and transformation of different types of land use patches [23,41]. The kappa coefficient (Equation (1)) is generally utilized to verify its spatial simulation accuracy, with a threshold range of [−1, 1], and when the value exceeds 0.6, the simulation results exhibit a high degree of consistency with the actual data [42].

$$\text{kappa} = \frac{(p_a - p_b)}{(1 - p_b)}, \tag{1}$$

where $p_a$ is the ratio of the correct simulation, and $p_b$ is the ratio of the expected simulation.

The system dynamics (SD) model is a simulation model of a complex system for ecological environmental and economic development proposed by J. W. Forrester in 1956, and it is powerful in terms of data simulation [43,44]. Two SD models, Model I (Figure 6) and Model II (Figure 7), were constructed in this study. Model I takes the trend of land use type change in Baishan as its main focus, and it contains three major subsystems: population, economy, and land use. Model II mainly constructs four subsystems (population, economy, land use, and environment) through a dynamic simulation of the type of construction land structure, the composition of the land investment, and the complex behavior of the system.

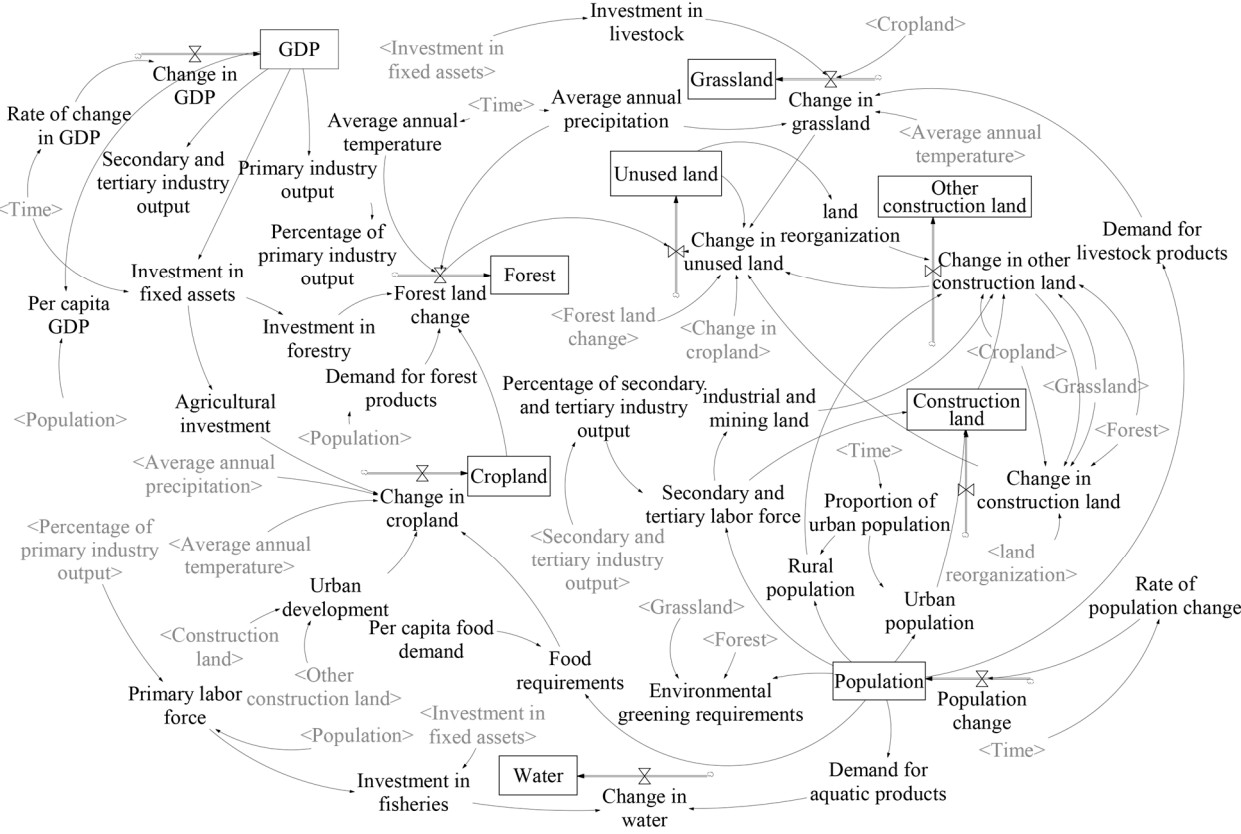

**Figure 6.** Flow chart of optimized allocation system of land use in Baishan (Model I). Note: Due to the far-off distance of some of the relevant variables in the figure, some of the variables were placed close together (i.e., gray text in the figure) for the sake of the overall coherence of the graph.

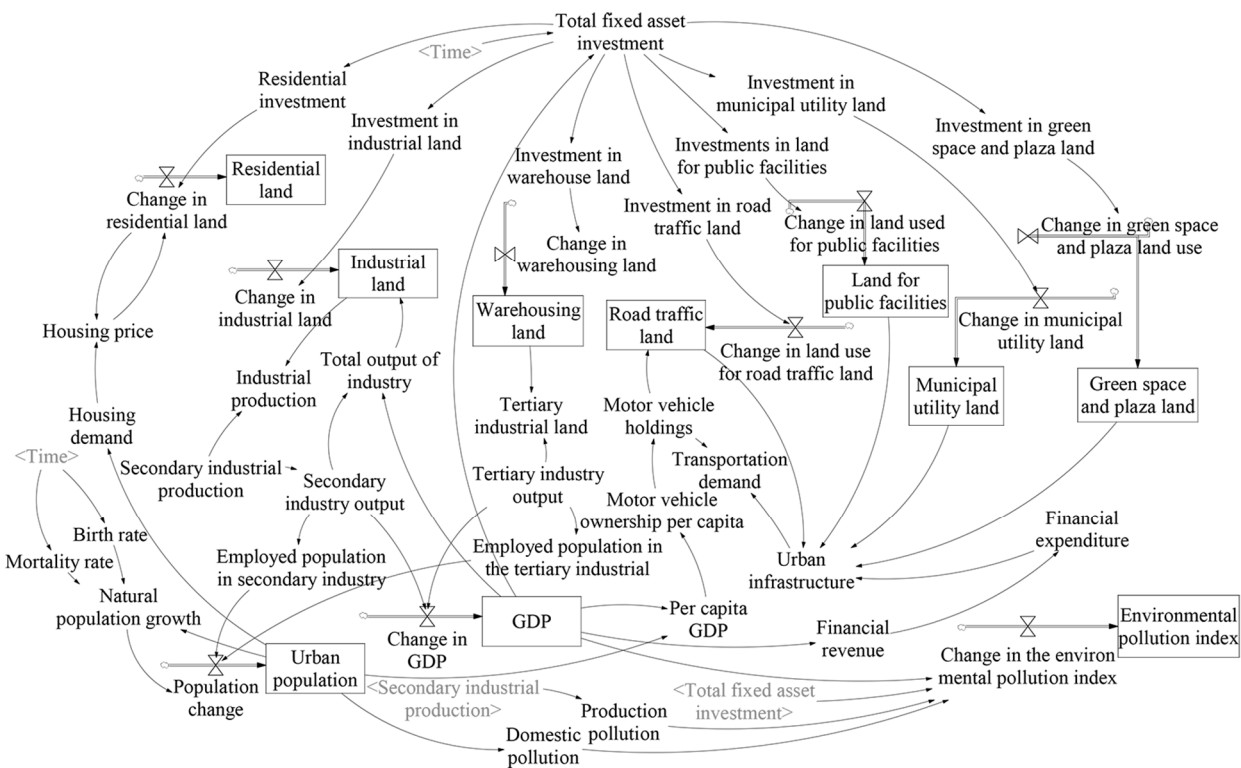

**Figure 7.** Flow chart of the optimized allocation system of construction land structure in Baishan (Model II).

This model usually utilizes the relative error (Equation (2)) of the historical simulation test to assess the scientific validity of the simulation findings, and the simulation test results must have an error of less than 5% [45].

$$\text{relative error} = \frac{|\text{simulated value} - \text{historical value}|}{\text{historical value}} \times 100\%, \tag{2}$$

The optimal allocation of construction land in shrinking cities involves multiple economic and social systems and is characterized by complexity and variability, rendering it challenging for a solitary model to reveal the fundamental mechanisms behind the alterations of construction land in shrinking cities [46,47]. Therefore, in this paper, we constructed a coupled PLUS-SD model with complementary functions, effectively leveraging the strengths of the SD model in simulating complex systems and the PLUS model in expressing spatial relationships. The working principle is shown in Figure 8.

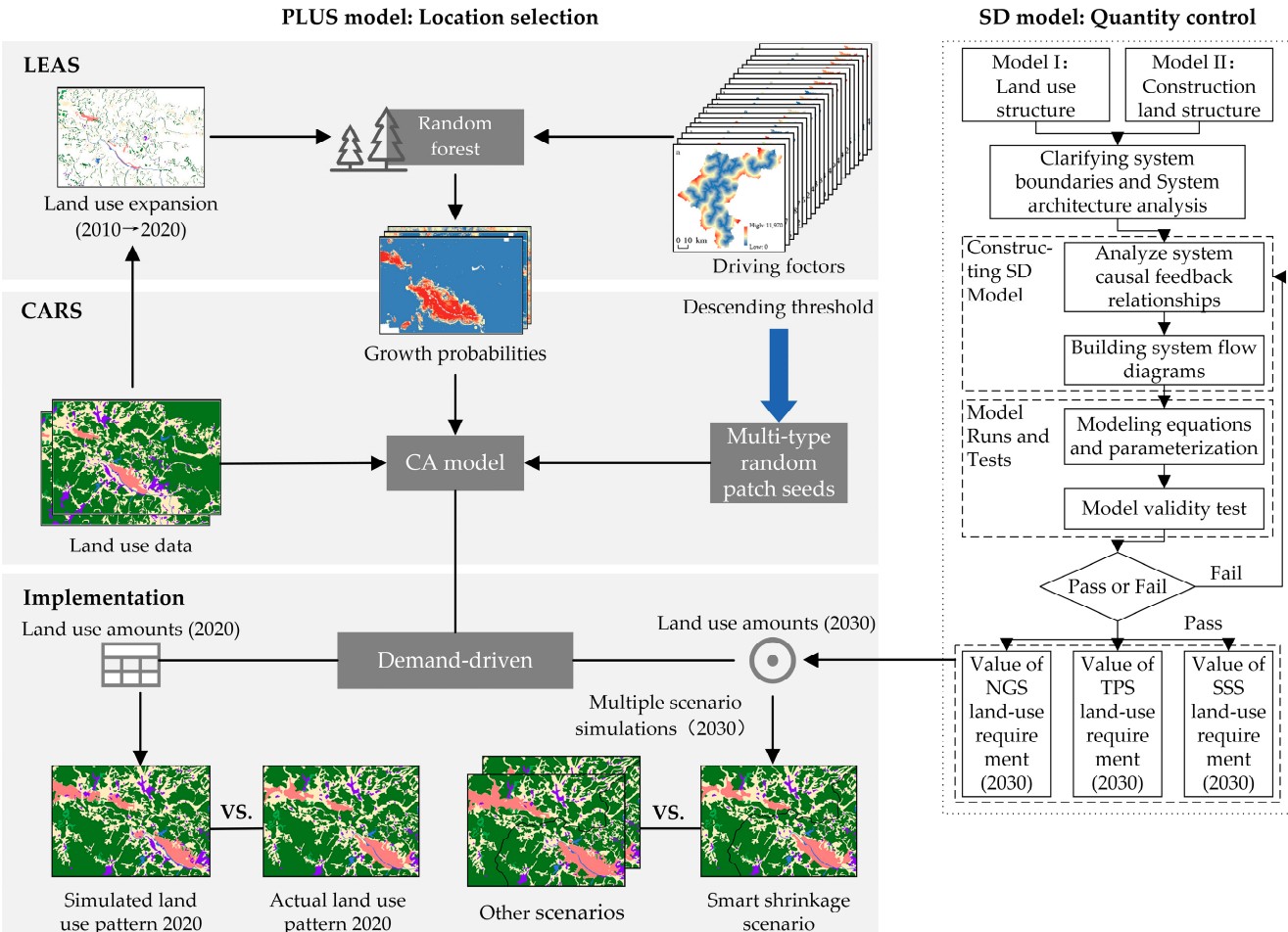

**Figure 8.** Construction and operation flowchart of the coupled PLUS-SD model.

The selection of the simulation scenarios is a reflection of the different future development plans and can lead to differences in land use change. Therefore, the following three scenarios were set up in this paper. (1) The natural growth scenario (NGS): The land use is not subject to any policy and planning constraints, and the land use and layout, as well as the structure of the construction land, change in accordance with historical inertia. This scenario is the base reference scenario and provides a frame of reference for the other two scenarios. (2) The traditional planning scenario (TPS): This scenario is a concrete manifestation of growthist development planning, emphasizing urban expansion with economic growth as the primary goal. Based on the planning objectives of the Baishan City Master Plan (2015–2030), the city's construction land and its structure are adjusted, and the

parameters of the neighborhood weights are changed accordingly. (3) The smart shrinkage scenario (SSS): This scenario reflects the requirements of the spatial development strategy aimed at enhancing the quality of the human living environment of inhabitants within a limited space. Based on the core connotations of smart shrinkage and the experience of planning practices, it relies on the per capita construction land and per capita residential land standards in China's Urban Land Use Classification and Planning and Construction Land Use Standards (GB50137-2011) [48], as well as the demand values of the various land use types in Baishan in 2030. The neighborhood weighting parameters are adjusted.

### 3.3.2. Compactness of Construction Land

Compactness can be used to express the shape characteristics of a city, as it is an indicator reflecting spatial intensification within an urban spatial form, and it has received extensive attention from scholars worldwide [49,50]. The circle is one of the most compact shapes and has a high degree of spatial aggregation among the parts contained within it. Therefore, the compactness formula defined in this paper is an evolution of the formula based on a circle. This formula is as follows [51]:

$$C = \frac{2\sqrt{\pi S}}{P}, \tag{3}$$

where $C$ is the compactness index; $S$ is the construction land area of the city; and $P$ is the perimeter of the city construction land. The threshold value of $C$ ranges from [0–1], and as the value of $C$ increases, the urban space becomes increasingly condensed.

### 3.3.3. Construction Land Intensification

According to existing research results [52,53], we selected 13 indicators from the four aspects of the input level, utilization intensity, output benefit, and sustainability to construct an evaluation index system for the intensive utilization of construction land (Table 4). Among them, the input level reflects the funds and materials used for city construction land, the utilization intensity characterizes the capacity of the city construction land to carry the population and economic activities, the output benefit mainly reflects the effect of the construction land use, and the sustainability reflects the social and environmental benefits of the intensive use.

**Table 4.** Evaluation index system and weighting of the intensive utilization of construction land.

| Standardized Layer | Index Level | Units | Index Attribute | Weights [1] |
|---|---|---|---|---|
| A Input level 0.2339 | $A_1$ Per capita urban road traffic area | $m^2/m^2$ | + | 0.0809 |
| | $A_2$ Per capita area of public utilities | $m^2/m^2$ | + | 0.0814 |
| | $A_3$ Per capita investment in fixed assets | Yuan/$m^2$ | + | 0.0716 |
| B Utilization intensity 0.2220 | $B_1$ Per capita construction land area | $m^2$/person | − | 0.0722 |
| | $B_2$ Per capita residential floor space | $m^2$/person | − | 0.0775 |
| | $B_3$ Per capita residential land area | $m^2$/person | − | 0.0723 |
| C Output benefits 0.3894 | $C_1$ Per capita secondary industry output | Yuan/$m^2$ | + | 0.0802 |
| | $C_2$ Per capita tertiary industry output | Yuan/$m^2$ | + | 0.0755 |
| | $C_3$ Per capita total retail sales of consumer goods | Yuan/$m^2$ | + | 0.0776 |
| | $C_4$ Per capita GDP output | Yuan/$m^2$ | + | 0.0768 |
| | $C_5$ Per capita fiscal revenue | Yuan/$m^2$ | + | 0.0793 |
| D Sustainability 0.1547 | $D_1$ Urban greening coverage rate | % | + | 0.0717 |
| | $D_2$ Green spaces and square area per capita | $m^2$/person | + | 0.0830 |

[1] The weights were computed using the entropy method.

In this paper, the entropy value method is applied to determine the weight of each indicator within the evaluation index system of the intensive utilization of construction land (Table 4). The specific calculation formulas are as follows:

$$S_{ij} = \frac{X_{ij} - minX_{ij}}{maxX_{ij} - minX_{ij}}, \tag{4}$$

$$W_{ij} = \frac{1 + \frac{1}{\ln n}\sum_{n=1}^{n}\left(\frac{S_{ij}}{\sum_{i=1}^{n}S_{ij}} \times \ln \frac{S_{ij}}{\sum_{i=1}^{n}S_{ij}}\right)}{\sum_{j=1}^{n}\left[1 + \frac{1}{\ln n}\sum_{n=1}^{n}\left(\frac{S_{ij}}{\sum_{i=1}^{n}S_{ij}} \times \ln \frac{S_{ij}}{\sum_{i=1}^{n}S_{ij}}\right)\right]}, \tag{5}$$

$$W_i = \sum_{j=1}^{n} W_{ij}, \tag{6}$$

$$I_i = \sum_{j=1}^{n}(W_{ij} \times S_{ij}), \tag{7}$$

where $S_{ij}$ and $X_{ij}$ are the standardized data value and original data value of the $j$th point of the $i$th standardiesd layer, $maxX_{ij}$ and $minX_{ij}$ are the maximum and minimum value of the indicator, $n$ is the number of indicators included in the indicator system, $W_{ij}$ is the weight of each indicator, $W_i$ is the weight of each standard layer, and $I_i$ is the degree of urban construction land intensification.

## 4. Results

### 4.1. Validation of the Accuracy of the Coupled PLUS-SD Model

Before analyzing the simulation results of the PLUS-SD model, it is imperative to validate the simulation efficacy of the SD model. The historical test time determined in this paper is 2020, and the test results are presented in Table 5. According to the findings, the relative errors of the historical tests of the two models are less than 5%, implying that the simulation accuracy of the two models is high and can effectively represent the causal feedback link between the variables in the system; it can also accurately predict the structure of the city land system in a scientifically valid manner.

**Table 5.** Historicity test table for the SD model simulation results in 2020.

| Model I | Cropland | Forest | Grassland | Water | CL [1] | UL [2] | OCL [3] |
|---|---|---|---|---|---|---|---|
| Historical value (km$^2$) | 465.03 | 2122.32 | 16.99 | 10.16 | 41.33 | 0.73 | 47.04 |
| Simulated value (km$^2$) | 465.64 | 2123.36 | 16.41 | 10.02 | 41.81 | 0.75 | 48.49 |
| Relative error (%) | 0.13 | 0.05 | 3.42 | 1.37 | 1.17 | 3.26 | 3.08 |
| **Model II** | **RL** | **PUL** | **IL** | **WL** | **RTL** | **MUL** | **GSPL** |
| Historical value (km$^2$) | 18.39 | 4.04 | 6.88 | 0.94 | 6.39 | 1.05 | 5.47 |
| Simulated value (km$^2$) | 18.58 | 4.07 | 6.99 | 0.94 | 6.39 | 1.04 | 5.67 |
| Relative error (%) | 1.01 | 0.63 | 1.63 | 0.03 | 0.05 | 0.51 | 3.57 |

[1] CL is construction land; [2] UL is unused land; [3] OCL is other construction land.

Subsequently, the kappa coefficients of the PLUS model and coupled PLUS-SD model in the 2020 land use simulation and the actual situation in 2020 were verified using 1% random sampling. The kappa coefficient of the PLUS model is 0.818, with an overall accuracy of 0.935, while that of the coupled PLUS-SD model is 0.822, with an overall accuracy of 0.937 (Figure 9). The coupled PLUS-SD model has a simulation performance superior to that of the PLUS model, and its simulation outcomes are more closely aligned with the actual progression of city land use.

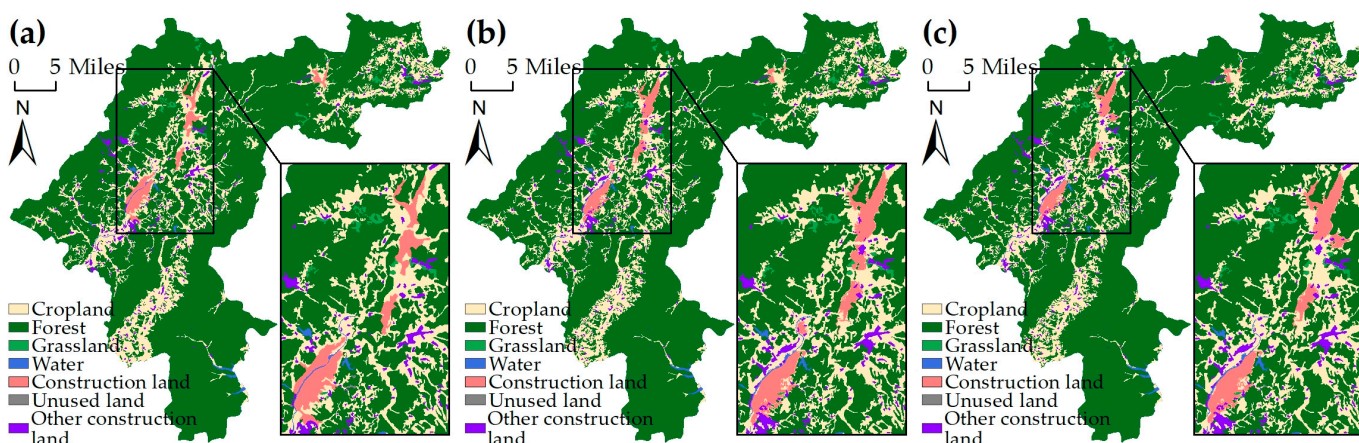

**Figure 9.** (**a**) Real distribution of land use types in Baishan in 2020; (**b**) Simulation results of land use types in Baishan in 2020 obtained using the PLUS model; (**c**) Simulation results of land use types in Baishan in 2020 obtained using the coupled PLUS-SD model.

*4.2. Characteristics of the Spatial Pattern of Construction Land under Different Scenarios*

In order to explore the development pattern of the construction land in Baishan under different development scenarios in 2030 and the spatial reconfiguration process of the land elements in different stages, in this paper, we focused on analyzing the conversion relationship between the construction land in Baishan and the other six land use types (Figure 10). In 2030, the areas with more changes in the construction land in the city districts in Baishan across all three simulation scenarios will be located at the edges of the landforms. The dominant expansion wing of the city's construction land extends along the southeast–northwest axis, with a significant trend of reciprocal expansion between the city districts (Table 6).

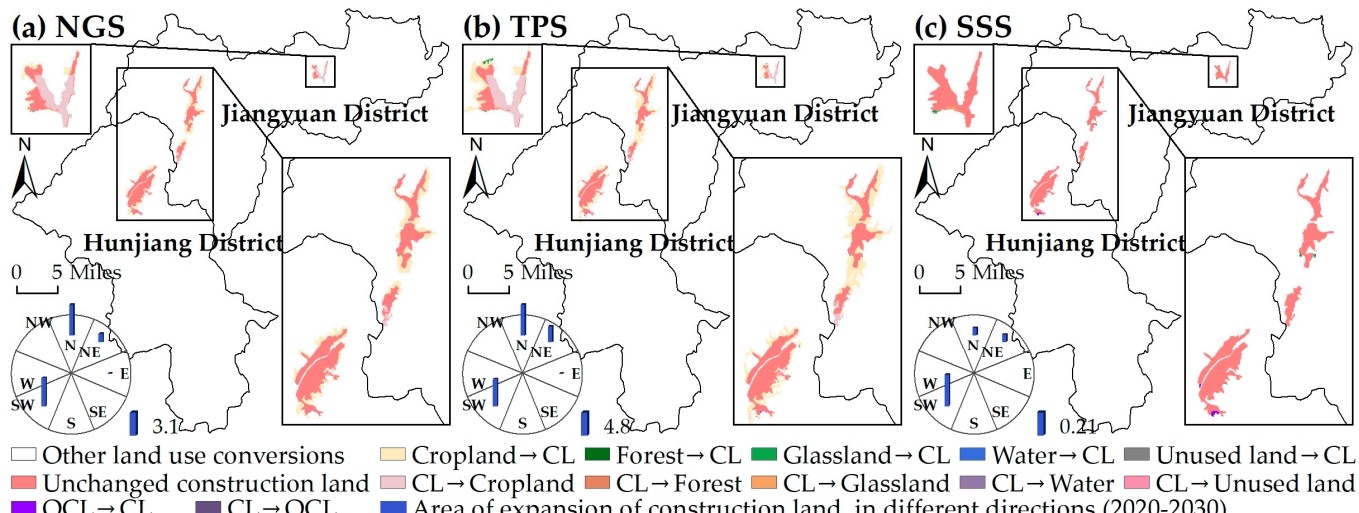

**Figure 10.** Spatial distribution of construction land conversion in Baishan under different scenarios during 2020–2030.

Under the NGS, the evolution of the spatial distribution of the construction land in the districts of Baishan in 2030 will continue the existing trend of change, which is mainly characterized by a substantial increase in the amount of construction land in the districts of the city (an increase of 30.72%). This is mainly the result of a reduction in cropland, particularly in the peri-urban regions, as well as a relatively large increase in the structure of the six categories of urban construction land, except for a slight reduction in the warehousing land. However, because Baishan has become a shrinking city and is likely

to remain so for an extended period of time, this scenario largely leads to the increasingly serious phenomenon of reverse polarization of the population loss and construction land expansion, the progressive expansion of unoccupied and deserted urban buildings, and a persistent decline in the quality of life of the city's population. This will have a negative impact on the city's population flow and will also lead to a vicious cycle of city shrinkage.

**Table 6.** Simulation results of construction land and its structure in Baishan under different scenarios.

| Year | | Construction Land (km²) | Land Structure for Urban Construction (km²) | | | | | | |
|---|---|---|---|---|---|---|---|---|---|
| | | | RL | PUL | IL | WL | RTL | MUL | GSPL |
| 2020 | | 41.33 | 18.39 | 4.04 | 6.88 | 0.94 | 6.39 | 1.05 | 5.47 |
| 2030 | NGS (km²) | 54.02 | 20.62 | 6.54 | 9.69 | 0.80 | 10.09 | 1.40 | 7.44 |
| | TPS (km²) | 62.98 | 16.30 | 6.52 | 19.41 | 2.55 | 10.73 | 1.43 | 7.58 |
| | SSS (km²) | 41.95 | 12.87 | 4.66 | 7.57 | 0.75 | 7.03 | 1.16 | 9.30 |

Under the TPS, the development of the quantity, scale, and spatial layout of the construction land and its structure, as well as the construction of major facilities in Baishan, is guided by the traditional urban development plan that prioritizes economic growth. Compared with the NGS, this scenario will lead to a greater expansion of the construction land in Baishan in 2030 (50.40% increase). In addition, to foster the swift growth of the city's economy, there will be substantial expansion in the size of the city's industrial and warehousing land area. This indicates the difficulty of traditional economic growth-oriented urban development planning to cope with the development realities of shrinking cities. During this period, because the NGS and TPS result in large increases in the area of the land for construction, the spatial structure of the city evolves from a multi-center cluster city formed by development zones and new districts to a ribbon city where the construction of new towns and new districts is integrated with the old city. However, this pattern of urban development at the expense of urban food security and encroachment on cropland will inevitably result in reduced urban food production and will pose a threat to food security. In addition, the urban–rural population shifts caused by urban economic growth can lead to a reduction in the quantity and quality of the food producer's labor force, which in turn causes a decline in the incentive to produce food. Therefore, if the shrinking city still implements the traditional urban development plan, it will aggravate the growth of the expansion speed and intensity of the city construction land and further deepen the contradiction of the city human–land relationship.

Under the guidance of the SSS concept, the spatial structure of the construction land in the districts of Baishan in 2030 still maintains a multi-center cluster structure, and the extent of the new construction land can be regulated, with a slight increase of only 0.62 km², resulting in the return of the already imbalanced human–land relationship to a balanced direction. To a certain extent, this will safeguard the city's food security. Throughout this time period, the area of residential land and warehousing land in the city will decline dramatically, while the area of the green spaces and plazas will increase dramatically, which means that the demolition and greening of abandoned and vacant buildings in the Baishan district under this scenario can contribute to the attainment of equilibrium between availability and need in the urban built environment. This can help optimize the structure of the city's construction land.

*4.3. Characteristics of Construction Land Use Efficiency under Different Scenarios*

In 2030, under all three simulation scenarios, the construction land in Baishan and its districts moves in the direction of increasing looseness, with a decreasing trend in compactness (Table 7). The reason for this is that Baishan is situated in the interior region of the Changbai Mountains, in the territory of mountain peaks, with the expansion of construction land occurring as a result of the topographic conditions and the size of the urban hinterland,

as well as the expansion of the construction land along the mountains and valleys; therefore, the shape tends to be irregular, leading to a decrease in the compactness of the city. In addition, compared to the Hunjiang District, the Jiangyuan District has decentralized development and a somewhat lower level of compactness due to the implementation of a one district and many parks management model. Based on the different simulation scenarios, the SSS has the smallest reduction in urban compactness in 2030 (0.0103).

**Table 7.** Changes in the compactness of the construction land in Baishan under different scenarios.

| Different Scenarios | | Name | Area (km²) | Perimeter (km) | Compactness | Tendency |
|---|---|---|---|---|---|---|
| 2020 | | Hunjiang district | 20.8980 | 76.62 | 0.2115 | / |
| | | Jiangyuan district | 20.4345 | 98.88 | 0.1621 | / |
| | | Baishan city | 41.3271 | 174.96 | 0.1303 | / |
| 2030 | NGS | Hunjiang district | 26.6031 | 125.94 | 0.1452 | non-compact |
| | | Jiangyuan district | 27.4248 | 185.82 | 0.0999 | non-compact |
| | | Baishan city | 54.0225 | 311.22 | 0.0837 | non-compact |
| | TPS | Hunjiang district | 29.2950 | 145.38 | 0.1320 | non-compact |
| | | Jiangyuan district | 33.6942 | 194.58 | 0.1058 | non-compact |
| | | Baishan city | 62.9838 | 339.48 | 0.0829 | non-compact |
| | SSS | Hunjiang district | 21.3201 | 82.02 | 0.1996 | non-compact |
| | | Jiangyuan district | 20.6307 | 109.80 | 0.1466 | non-compact |
| | | Baishan city | 41.9463 | 191.28 | 0.1200 | non-compact |

In 2030, the expansion of construction land in Baishan under the SSS tends to be in the direction of intensification, with an increase of 0.0158, which is in line with the analytical idea of an escalation in the level of intense utilization of city construction land under the guidance of the smart shrinkage concept (Table 8). Under this scenario, the growth of construction land, especially residential land, can be controlled, which enhances the city's population agglomeration capacity and the intensity of construction land use. Furthermore, this scenario prioritizes the sustainable development of the city, as well as the enhancement of the urban environment's quality and ecological livability. The shrinkage of the city will increase the potential of the city to re-create itself by remodeling idle and abandoned construction land and increasing the coverage of urban green space, as well as by expanding the area of green plazas and squares, to improve the sustainability of the city's construction land. In addition, construction land intensification decreases by 0.0346 and 0.0532 under the NGS and TPS, respectively. This is in line with the logic of lowering the level of the intensive utilization of construction land due to the lower intensity and output efficiency of urban construction land use as a result of the exodus of the urban population and the shrinking of socio-economic activities under the development plan that puts economic development as the primary objective.

**Table 8.** Statistics of the level of intensive utilization of construction land in Baishan under different scenarios.

| Different Scenarios | | Intensity | Input Level | Utilization Intensity | Output Benefits | Sustainability |
|---|---|---|---|---|---|---|
| 2020 | | 0.3001 | 0.0587 | 0.1011 | 0.0775 | 0.0628 |
| 2030 | NGS | 0.2655 | 0.0563 | 0.0833 | 0.0584 | 0.0674 |
| | TPS | 0.2469 | 0.0519 | 0.0781 | 0.0492 | 0.0677 |
| | SSS | 0.3159 | 0.0598 | 0.1073 | 0.0771 | 0.0718 |

In summary, compared with the NGS (which continues the status quo of low-density urban development) and the TPS (growth-oriented), the SSS can curb problems in land use, such as the disorderly expansion and inefficient use of construction land, an increase in idle and abandoned construction land, and an irrational construction land structure, all of which are caused by Baishan's shrinkage process. It can also reverse the new contradictory

development pattern of the co-occurrence of population loss and the expansion of construction land in the city, which will help to optimize the quantitative, hierarchical, and effective structure of Baishan's land resources. In addition, it is a scientific and feasible plan for the realization of the efficient and sustainable use of construction land in Baishan.

## 5. Discussion

The substantial differences in the national development backgrounds and histories, as well as the institutional environments, make it impractical to copy developed countries' responses to shrinking cities [2], but we can still explore ways of coping with shrinking cities that are adapted to China's national conditions by borrowing the concept of smart shrinkage planning. During the 40 years of rapid economic development since China's reform and opening up, no city has been willing to voluntarily recognize and accept the reality of urban shrinkage. This has resulted in a discourse of local development that still follows the planning paradigm and inertia of the traditional planning scenario, in which preserving economic growth is still largely regarded as a justification for expansive investment and sprawling development. However, in the context of population contraction, the implementation of economic growth-oriented urban development planning in shrinking cities, such as Baishan, under the NGS and TPS, will lead to a significant expansion of urban construction space, and the operational efficiency of the city's infrastructure, such as the water supply, electricity supply, and public transportation, will inevitably be reduced as the user count decreases, the service area increases, and the costs of construction and maintenance increase [54]. In addition, it will also lead to a sharp increase in the quantity of vacant and unused buildings in the city, which often seriously negatively affects the urban real estate market and aggravates the conflict between developers and tenants [55]. Moreover, it often provides free places for low-income people, the unemployed, and even criminals in the city, raising the crime rate in the neighborhood [56]. The collapse of the real estate market and the rise of crime in the neighborhood could further lead to the displacement of the city's current residents, creating a vicious cycle of city shrinkage.

However, under the guidance of the concept of smart shrinkage planning, such as the city of Baishan in 2030 under the SSS, the scale of new construction land in the city can be controlled, the area of residential land in the city can be reduced, and the population of the low-density areas of the city can be relocated to high-density areas [57], which can induce the population of the city to gather, improve the level of comprehensive services of the old urban areas of the city, improve the problem of hollowing out from the living space, and improve the quality of life of the residents of the old urban areas by further improving the public service facilities of the old urban areas and the municipal public facilities. Furthermore, removing unused and abandoned buildings and public facilities in low-density areas can reduce local government financial expenditures and improve public security and the public environment in the city by transforming the sites of unused and abandoned buildings into industrial art parks [58].

## 6. Conclusions

As the population of shrinking cities progressively decreases, the traditional growth model of planning and land development has created a paradoxical phenomenon of population loss and expansion of construction land, which has, in turn, led to problems such as urban ecological damage, public safety crises, the inefficient operation of public service facilities, and the collapse of the real estate market. Unlike original growth-oriented planning, some shrinking cities in the United States have introduced smart shrinkage-oriented planning in recent years after having accepted the fact that the city's population is shrinking and have reduced incremental planning so as to achieve a balance of the relationship between people and the land. In this study, Baishan, a typical shrinking city with serious population outflow, continuous expansion of construction land, and inefficient land utilization, was chosen as a case study. By constructing a coupled PLUS-SD model, the spatial pattern and utilization efficiency differences of the city's construction land under the

three development scenarios of natural growth, traditional planning, and smart shrinkage were simulated, after which the feasibility of the implementation of the smart shrinkage theory in China's shrinking urban planning was examined. The results of this study provide a reference for stopping the deterioration of the population–land relationship in shrinking cities and provide a basis for the formulation of development plans for shrinking cities in China. The main conclusions of this study are as follows.

(1)  The coupled PLUS-SD model, which was improved in this study, has a kappa coefficient of 0.822 and an overall accuracy of 0.937, which are better than those of the PLUS model. According to the simulation results, under the three scenarios, the areas with the most changes in construction land in Baishan will be mostly located at the edges of the landforms in 2030. The dominant expansion wing of the city's construction land will extend along the southeast–northwest axis, with a significant trend of reciprocal expansion between city districts.

(2)  Compared with the natural growth scenario, which perpetuates the status quo of low-density urban development, the growth-oriented traditional planning scenario will increase the area of expansion of the construction land in Baishan. In both scenarios, the utilization intensity and output benefits will shrink, leading to a decrease in the level of the intensive use of urban construction land as a result of population outflow and shrinking socio-economic activities.

(3)  The smart shrinkage scenario shifts the focus of urban development to the redevelopment of existing land stock and optimization of the urban construction land structure in order to promote the development of the city in a healthy and sustainable direction. In this scenario, an increase in the utilization intensity and sustainability of urban construction land use by strictly controlling the scale of new urban construction land and improving the quality of the urban environment and ecological livability leads to an increase in the level of intensive utilization of urban construction land.

(4)  The concept of smart shrinkage planning can solve the problems of the disorderly expansion of urban construction land, the increase in idle construction land, the irrational structure of construction land, and the polarization of the human–land relationship. In addition, it can help to optimize the quantitative structure, hierarchical structure, and benefit structure of the land resources in shrinking cities. It is a scientific and feasible plan for the realization of efficient and sustainable utilization of the construction land in shrinking cities.

Nonetheless, this study has the following limitations. First, because of the lack of spatial data regarding construction land structure in Baishan, the coupled PLUS-SD model can only simulate the spatial distribution of the land use in Baishan under different scenarios, and it lacks optimization of the spatial layout of the construction land structure. Second, the governance of shrinking cities is a multi-dimensional and multi-scale worldwide problem, but it is a new topic for China and provides a new perspective for investigating the development of urbanization. How to scientifically and rationally guide China's urbanization and avoid the disorderly development and spreading of the urban fringe is not only a difficult urban development problem faced by the current government but also an important direction for the authors' future research.

**Author Contributions:** Conceptualization, W.L., H.L. and S.W.; Methodology, Z.F.; Software, W.L.; Validation, W.L., F.H. and S.W.; Formal analysis, W.L.; Investigation, W.L.; Resources, S.W.; Data curation, W.L.; Writing—original draft, W.L.; Writing—review & editing, H.L., F.H., Z.F. and S.W.; Visualization, F.H.; Supervision, H.L.; Project administration, W.L.; Funding acquisition, F.H. and S.W. All authors have read and agreed to the published version of the manuscript.

**Funding:** This research was funded by the National Natural Science Foundation of China (No. 42171198; No. 42171191; No. 42071219).

**Data Availability Statement:** Data is contained within the article.

**Conflicts of Interest:** The authors declare no conflict of interest.

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
