# Peer review of "A Simulation of the Spatial Expansion Process of Shrinking Cities Based on the Concept of Smart Shrinkage: A Case Study of the City of Baishan"

_land, doi:10.3390/land13020239_

Round 1

Reviewer 1 Report

Comments and Suggestions for Authors

Smart shrinkage planning is a scientific  and  feasible  plan for realizing the  efficient  and  sustainable  use  of  construction  land in shrinking cities, it’s essential to study reasonable methods to plan shrinking city. The research objectives are clear and the methods used in the research are suitable to answer the research questions. The results are sense for planning sustainable city. Some minors need to be done before published.

1)  Figure 3 need to be revised with northeast in China, Baishan in Northeast China.

2)  In section 3.2, where are the spatial difference among the three scenarios need to be discussed.

3)  In Section 4.1, appearance of smart Shrinkage in Youngstown is not suitable. Please put them into the discussion within suitable location. What you want to study is Smart Shrinkage in Baishan and Smart Shrinkage in Youngstown is just a reference.you want to study is Smart Shrinkage in Baishan and Smart Shrinkage in Youngstown is just a reference.the 天河 smart shrinkageplanningis a 24scientific and feasible plan for realizingthe efficient and sustainable use of construction landin 25shrinking cities

smart shrinkageplanningis a 24scientific and feasible plan for realizingthe efficient and sustainable use of construction landin 25shrinkin

Comments on the Quality of English Language

Smart shrinkage planning is a scientific  and  feasible  plan for realizing the  efficient  and  sustainable  use  of  construction  land in shrinking cities, it’s essential to study reasonable methods to plan shrinking city. The research objectives are clear and the methods used in the research are suitable to answer the research questions. The results are sense for planning sustainable city. Some minors need to be done before published.

1)  Figure 3 need to be revised with northeast in China, Baishan in Northeast China.

2)  In section 3.2, where are the spatial difference among the three scenarios need to be discussed.

3)  In Section 4.1, appearance of smart Shrinkage in Youngstown is not suitable. Please put them into the discussion within suitable location. What you want to study is Smart Shrinkage in Baishan and Smart Shrinkage in Youngstown is just a reference.you want to study is Smart Shrinkage in Baishan and Smart Shrinkage in Youngstown is just a reference.the 天河 smart shrinkageplanningis a 24scientific and feasible plan for realizingthe efficient and sustainable use of construction landin 25shrinking cities

Author Response

Dear reviewer,

The authors would like to thank you for the patient review and the affirmative comments of our manuscript (Manuscript ID: land-2844540; Title: Simulation of the Spatial Expansion Process of Shrinking Cities Based on the Concept of Smart Shrinkage: A Case Study of Baishan City). There were three reviewers who had made comments on our manuscript, and we were offered only ten days to revise the paper and have it checked for linguistic errors. The time is a little bit pressed for the revision according to these many comments. However, the authors have studied your comments very carefully and revised the manuscript strictly in accordance with your suggestion. The revisions and explanation were listed as follow.

*The paper has been edited by a native English-speaker with a higher degree in a relevant discipline.

*Hope the revision meets requirements for publication and please let us know if you have further comments.

Sincerely yours,

Wancong Li, Hong Li, Feilong Hao, Zhiqiang Feng and Shijun Wang

Reviewer 2 Report

Comments and Suggestions for Authors

This manuscript presents a study on "Simulation of the Spatial Expansion Process of Shrinking Cities Based on the Concept of Smart Shrinkage: A Case Study of Baishan City." The paper is interesting. However, some inconsistencies within the manuscript that should be clarified before acceptance for publication can be recommended.

The most important thing is the arrangement of the manuscript. It hampers understanding your research concept. I suggest re-adjusting some parts of the manuscript.

Comments -

1 What is the originality of this study? Please emphasize the end of the introduction section.

2. Research gap should be enhanced.

3. What is the interaction between Figure 2. and research methodology?

4. Compress the text of the study area.

5. Figures 5 and 6 should be clearly interpreted before the figures. As well as, what are the meaning of the bold color text and gray color text. Please mention this in the figure caption.

6. Figures 5 and 6 have the same figure caption, so what is the difference between these two figures?

7. The methodology section should be explained clearly. This study used 2000, 2010, and 2020 raster data. Driving variables (as mentioned in Table 3) come from different years and resolutions. How do authors combine these for your model? A scientific explanation is needed.

8. compress the conclusion section.

9. English should be checked.

Comments on the Quality of English Language

Moderate editing of English language required.

Author Response

Dear reviewer,

Thank you so much for your conscientious and insightful review on our manuscript (Manuscript ID: land-2844540; Title: Simulation of the Spatial Expansion Process of Shrinking Cities Based on the Concept of Smart Shrinkage: A Case Study of Baishan City). The authors had learned a lot and improved our research work following your comments and suggestions. There were three reviewers who had made comments on our manuscript, and we were offered only ten days to revise the paper and have it checked for linguistic errors. However, we did have studied your comments very carefully and made revisions according to constructive suggestions.

*The paper has been edited by a native English-speaker with a higher degree in a relevant discipline.

*Hope the revision meets requirements for publication and please let us know if you have further comments.

Sincerely yours,

Wancong Li, Hong Li, Feilong Hao, Zhiqiang Feng and Shijun Wang

Reviewer 3 Report

Comments and Suggestions for Authors

A study on the e feasibility of  smart shrinkage planning in shrinking cities in building  a coupled PLUS-SD model is presented.

In the introductory section the authors must highlight the research goals and innovative aspects more than related research proposed in recent literature.

The framework in Fig. 2 needs to be discussed in more detail. It should be presented in a discussion session of the proposed methodology rather than in the introductory section.

The discussion of the flowchart of the coupled PLUS-SD model shown in Fig. 7 needs to be broader. In particular, the functionalities of the SD model and the PLUS model need to be briefly described.

It is not clear how the four atstandardized layers in Tab. 4 are constructed. First, the units of measurement of the indices are different and it is not clear how they are used with their weights to determine the standardized layer; furthermore it is not clear how the weights are obtained; in fact, it is not enough to say that the entropy method is used, but it is necessary to specify which entropy method was used for the weight assessment.

A brief discussion on the future prospects of research should be included in the final section.

Comments on the Quality of English Language

Some grammar typos are present in the text; they must be removed.

Author Response

Dear reviewer,

Sincere thanks to you for the patient review and affirmative comments of our manuscript. We really appreciate your constructive comments and we’ve been aware of the shortcomings of our manuscripts (Manuscript ID: land-2844540; Title: Simulation of the Spatial Expansion Process of Shrinking Cities Based on the Concept of Smart Shrinkage: A Case Study of Baishan City). There were three reviewers who had made comments on our manuscript, and we were offered only ten days to revise the paper and have it checked for linguistic errors. However, the authors have studied your comments carefully and revised the manuscript as follow.

*The paper has been edited by a native English-speaker with a higher degree in a relevant discipline.

*Hope the revision meets requirements for publication and please let us know if you have further comments.

Sincerely yours,

Wancong Li, Hong Li, Feilong Hao, Zhiqiang Feng and Shijun Wang

Round 2

Reviewer 2 Report

Comments and Suggestions for Authors

I accept the current version of the manuscript.

Reviewer 3 Report

Comments and Suggestions for Authors

The authors have revised their manuscript taking into account all my suggestions and improving the quality of the document. I consider this paper publishable in the current version.